# sTREM-1 Predicts Disease Severity and Mortality in COVID-19 Patients: Involvement of Peripheral Blood Leukocytes and MMP-8 Activity

**DOI:** 10.3390/v13122521

**Published:** 2021-12-15

**Authors:** Pedro V. da Silva-Neto, Jonatan C. S. de Carvalho, Vinícius E. Pimentel, Malena M. Pérez, Diana M. Toro, Thais F. C. Fraga-Silva, Carlos A. Fuzo, Camilla N. S. Oliveira, Lilian C. Rodrigues, Jamille G. M. Argolo, Ingryd Carmona-Garcia, Nicola T. Neto, Camila O. S. Souza, Talita M. Fernandes, Victor A. F. Bastos, Augusto M. Degiovani, Leticia F. Constant, Fátima M. Ostini, Marley R. Feitosa, Rogerio S. Parra, Fernando C. Vilar, Gilberto G. Gaspar, José J. R. da Rocha, Omar Feres, Fabiani G. Frantz, Raquel F. Gerlach, Sandra R. Maruyama, Elisa M. S. Russo, Angelina L. Viana, Ana P. M. Fernandes, Isabel K. F. M. Santos, Vânia L. D. Bonato, Antonio L. Boechat, Adriana Malheiro, Ruxana T. Sadikot, Marcelo Dias-Baruffi, Cristina R. B. Cardoso, Lúcia H. Faccioli, Carlos A. Sorgi

**Affiliations:** 1Departamento de Análises Clínicas, Toxicológicas e Bromatológicas, Faculdade de Ciências Farmacêuticas de Ribeirão Preto—FCFRP, Universidade de São Paulo—USP, Ribeirão Preto 14040-903, SP, Brazil; pedrovieira@usp.br (P.V.d.S.-N.); jonatancarvalho@usp.br (J.C.S.d.C.); viniciuspimentel@usp.br (V.E.P.); malenac@usp.br (M.M.P.); dianamota.t@gmail.com (D.M.T.); cafuzo@usp.br (C.A.F.); camilla.narjara@outlook.com (C.N.S.O.); liktaldi@fcfrp.usp.br (L.C.R.); camila.oliveirasilva@usp.br (C.O.S.S.); victor_alx@usp.br (V.A.F.B.); frantz@fcfrp.usp.br (F.G.F.); elisa@fcfrp.usp.br (E.M.S.R.); mdbaruff@fcfrp.usp.br (M.D.-B.); cristina@fcfrp.usp.br (C.R.B.C.); 2Programa de Pós-Graduação em Imunologia Básica e Aplicada—PPGIBA, Instituto de Ciências Biológicas, Universidade Federal do Amazonas—UFAM, Manaus 69080-900, AM, Brazil; antonioluiz.boechat@gmail.com (A.L.B.); malheiroadriana@yahoo.com.br (A.M.); 3Departamento de Química. Faculdade de Filosofia, Ciências e Letras de Ribeirão Preto—FFCLRP, Universidade de São Paulo—USP, Ribeirão Preto 14040-901, SP, Brazil; 4Departamento de Bioquímica e Imunologia. Faculdade de Medicina de Ribeirão Preto—FMRP, Universidade de São Paulo—USP, Ribeirão Preto 14040-900, SP, Brazil; thaisfragasilva@gmail.com (T.F.C.F.-S.); imsantos@fmrp.usp.br (I.K.F.M.S.); vlbonato@fmrp.usp.br (V.L.D.B.); 5Escola de Enfermagem de Ribeirão Preto—EERP, Universidade de São Paulo—USP, Ribeirão Preto 14040-902, SP, Brazil; jamilleargolo@usp.br (J.G.M.A.); ingryd.garcia@usp.br (I.C.-G.); nicola.neto@usp.br (N.T.N.); talitafernandes@usp.br (T.M.F.); angelina.lettiere@usp.br (A.L.V.); anapaula@eerp.usp.br (A.P.M.F.); 6Hospital Santa Casa de Misericórdia de Ribeirão Preto, Ribeirão Preto 14085-000, SP, Brazil; augustomd@msn.com (A.M.D.); lelemed2008@hotmail.com (L.F.C.); tata_ostini@hotmail.com (F.M.O.); 7Departamento de Cirurgia e Anatomia, Faculdade de Medicina de Ribeirão Preto—FMRP, Universidade de São Paulo (USP), Ribeirão Preto 14048-900, SP, Brazil; mrfeitosa@hcrp.usp.br (M.R.F.); rsparra@hcrp.usp.br (R.S.P.); ribeiro.rocha0717@gmail.com (J.J.R.d.R.); omar.feres2021@gmail.com (O.F.); 8Departamento de Clínica Médica, Faculdade de Medicina de Ribeirão Preto—FMRP, Universidade de São Paulo (USP), Ribeirão Preto 14049-900, SP, Brazil; fcvilar@gmail.com (F.C.V.); ggaspar@hcrp.usp.br (G.G.G.); 9Departamento de Morfologia, Fisiologia e Patologia básica, Faculdade de Odontologia de Ribeirão Preto, Universidade de São Paulo (USP), Ribeirão Preto 14040-904, SP, Brazil; rfgerlach@forp.usp.br; 10Centro de Ciências Biológicas e da Saúde, Departamento de Genética e Evolução, Universidade Federal de São Carlos (UFSCar), São Carlos 13565-905, SP, Brazil; srmaruyama@gmail.com; 11Department of Internal Medicine, Division of Pulmonary, Critical Care and Sleep, College of Medicine, University of Nebraska Medical Center, Omaha, NE 68198, USA; rsadikot@unmc.edu

**Keywords:** COVID-19, sTREM-1, biomarker, inflammation, MMP-8

## Abstract

Uncontrolled inflammatory responses play a critical role in coronavirus disease (COVID-19). In this context, because the triggering-receptor expressed on myeloid cells-1 (TREM-1) is considered an intrinsic amplifier of inflammatory signals, this study investigated the role of soluble TREM-1 (sTREM-1) as a biomarker of the severity and mortality of COVID-19. Based on their clinical scores, we enrolled COVID-19 positive patients (*n* = 237) classified into mild, moderate, severe, and critical groups. Clinical data and patient characteristics were obtained from medical records, and their plasma inflammatory mediator profiles were evaluated with immunoassays. Plasma levels of sTREM-1 were significantly higher among patients with severe disease compared to all other groups. Additionally, levels of sTREM-1 showed a significant positive correlation with other inflammatory parameters, such as IL-6, IL-10, IL-8, and neutrophil counts, and a significant negative correlation was observed with lymphocyte counts. Most interestingly, sTREM-1 was found to be a strong predictive biomarker of the severity of COVID-19 and was related to the worst outcome and death. Systemic levels of sTREM-1 were significantly correlated with the expression of matrix metalloproteinases (MMP)-8, which can release TREM-1 from the surface of peripheral blood cells. Our findings indicated that quantification of sTREM-1 could be used as a predictive tool for disease outcome, thus improving the timing of clinical and pharmacological interventions in patients with COVID-19.

## 1. Introduction

Coronavirus disease (COVID-19), which is caused by the severe acute respiratory syndrome coronavirus 2 (SARS-CoV-2), became a global public health problem due to the important frequencies of mortality. Approximately 80% of patients infected with SARS-CoV-2 will exhibit mild symptoms or remain asymptomatic. However, this infection may lead to serious clinical conditions, such as acute respiratory distress syndrome (ARDS), cardiovascular disorders, coagulopathy, and shock, which can result in multiorgan system dysfunction in certain patients [1,2,3]. Elderly, male, and obese patients, or those with chronic comorbidities, are more likely to develop the worst outcomes [4]. Moreover, patients with COVID-19 were found to exhibit multiple hematological abnormalities, including lymphopenia and thrombocytopenia as the most prominent manifestations, in addition to developing neutrophilia [5].

Similarly to SARS, inflammation plays an important role in the pathophysiology of COVID-19, and patients with severe disease may have high serum concentrations of inflammatory markers such as IL-6, TNF, C-reactive protein (CRP), and D-dimer [6,7]. Triggering receptor expressed on myeloid cells 1 (TREM-1) is a member of the immunoglobulin superfamily expressed on myeloid and epithelial cells. Activation of TREM-1 induces the increased secretion of TNF-α, IL-6, IL-1β, IL-2, and IL-12p40 by monocytes, macrophages, and dendritic cells, which enhances inflammation during infection caused by different pathogens, such as influenza A virus, dengue virus, hepatitis C virus, *Plasmodium falciparum*, *Staphylococcus aureus*, and *Pseudomonas aeruginosa* [8,9,10,11]. The immune responses to viral and bacterial infections are modulated by the activation of TREM-1 in macrophages [12]. Furthermore, infection with hepatitis C virus and the human immunodeficiency virus (HIV), or even exposure to the HIV proteins Tat or gp120 without infection [13], induces the expression of TREM-1. The soluble form of TREM-1 (sTREM-1) has been detected in biological samples, such as plasma, during infection and inflammation processes [14,15]. sTREM-1 is a 27 kDa polypeptide consisting of the extracellular domain of TREM-1, which is released from the cell surface by metalloproteinase [16]. Furthermore, sTREM-1 is an important prognostic marker in diseases that also develop severe inflammatory pathways, such as sepsis and AIDS [10,15,17,18]. Recently, our group [19] and other works [20,21] suggested that sTREM-1 levels are related to COVID-19 severity and may serve as early triage tools for patients with adverse outcomes.

Given that SARS-CoV-2 induces a hyper-inflammatory response, we hypothesized that in COVID-19 patients, the virus could trigger activation and up-regulation of TREM-1 in the cell surface and, subsequently, shedding in the soluble form (sTREM-1) by metalloproteinase activity. Therefore, we conducted a prospective study to investigate the usefulness of sTREM-1 as a biomarker that can predict severity and mortality in COVID-19, in addition to confirming its risk stratification performance, thus guiding the development of effective interventions in patients who require intensive care after hospitalization.

## 2. Materials and Methods

### 2.1. Study Design and Participants

This prospective study was conducted at the Hospital Santa Casa de Misericórdia of Ribeirão Preto and Hospital São Paulo of Ribeirão Preto, Brazil from June to December of 2020, using stringent and reasonable inclusion and exclusion criteria: adults who tested positive for COVID-19 and controls (healthy volunteers) who tested negative for COVID-19; and exclusion for children under 18 years of age, and pregnant or lactating women. In total, 50 control subjects were included, along with 237 patients who tested positive for COVID-19, as determined by analyzing nasopharyngeal swabs using a genomic RNA assay with RT-PCR (Biomol OneStep Kit/COVID-19-Instituto de Molecular Biology of Paraná-IBMP Curitiba/PR, Brazil) or using serology-specific IgM and IgG antibodies tests (SARS-CoV-2 antibody test^®^, Guangzhou Wondfo Biotech Co., Ltd., Guangzhou, China).

### 2.2. Study Design and Participants

The data were collected from the electronic medical record systems. We included socio-demographic information, comorbidities, medical history, clinical symptoms, routine laboratory tests, immunological tests, chest computed tomography scan results, clinical interventions, and outcomes. Data collection from laboratory results was defined by considering the first examination at admission (within 24 h of admission) with an estimate of 6.1 ± 2.8 (mean ± SD) days after the onset of the symptom as the primary endpoint and the clinical outcome (death or recovery) as the secondary endpoint.

### 2.3. Severity Assessment

For assessment of clinical severity, patients with COVID-19 were classified into mild, moderate, severe, and critical groups, based on the modified statement in the Novel Coronavirus Pneumonia Diagnosis and Treatment Guideline (7th ed.) [22,23], shown in Appendix A.

### 2.4. Laboratory Methods

Blood samples were collected by venipuncture in tubes with a vacuum collection system. Two tubes (5 mL capacity) were collected from each patient: one tube containing EDTA anticoagulant (BD Vacutainer^®^ EDTA K2, Franklin Lakes, NJ, USA) to perform hematological tests, and a tube containing heparin anticoagulant (BD SST^®^ Gel Advance^®^, Franklin Lakes, NJ, USA) to obtain plasma used to quantify levels of circulating protein mediators by flow cytometry or ELISA assay. The samples were stored in a −80 °C freezer.

### 2.5. Cytokine Measurements

The levels of cytokines IL-1β, IL-6, IL-8, IL-10, IL-12, and TNF were measured using a Cytometric Bead Array kit according to the manufacturer’s specifications (BD^TM^ Human Inflammatory Cytokine CBA Kit, Catalog No. 551811, Lot: 9341655). The detection range of each cytokine was 5 to 5000 pg/mL. Data were acquired using a FACSCanto II flow cytometer and FACSDiva software (BD Biosciences, Franklin Lakes, NJ, USA). Data are presented as mean fluorescence intensity for each serum cytokine.

### 2.6. Detection of Plasma sTREM-1 and MMP-8

Systemic levels of sTREM-1 and matrix metalloproteinases (MMP)-8 were measured in plasma using an ELISA kit (DuoSet-Human TREM-1 and DuoSet-Human Total MMP-8, R&D System, Minneapolis, MN, USA) according to the manufacturer’s specifications.

### 2.7. Analysis of TREM-1-Positive Cells

Expression of TREM-1 in peripheral blood cells was determined by flow cytometry. Blood (1 mL) was used, and red blood cells were lysed using RBC lysis buffer (Roche Diagnostics GmbH, Mannheim, GR). Leukocytes were washed in PBS containing 5% fetal bovine serum (FBS) (Gibco™, Thermo Fisher Scientific, Waltham, MA, USA), centrifuged, and resuspended in Hank’s balanced salt solution (Sigma-Aldrich, Merck, Darmstadt, Germany) containing 5% FBS, followed by surface antigen staining. Briefly, cells were stained with fixable viability stain (1:1000) (BD Biosciences, San Diego, CA, USA) and incubated with monoclonal antibodies specific for CD14 (1:100) (M5E2; Biolegend, San Diego, CA USA), CD16 (1:100) (BV510-3G8; Biolegend, San Diego, CA USA), and TREM-1 (1:100) (FAB1278P; R&D Systems, Minneapolis, MN, USA) for 30 min at 4 °C. The stained cells were washed and fixed with BD Cytofix™ fixation buffer (554655; BD Biosciences, San Diego, CA, USA). Data acquisition was performed using the Fortessa™ LSR flow cytometer (BD Biosciences, San Jose, CA, USA) and the FACS-Diva software (version 8.0.1) (BD Biosciences, Franklin Lakes, NJ, USA). For the analysis, 100,000 events were acquired for each sample. CD14 and CD16 positive cells were gated, and TREM-1 expression analysis was performed using FlowJo^®^ software (version 10.7.0-Tree Star, Ashland, OR, USA) (Appendix A).

### 2.8. Statistical Analysis

Data are presented in tables and graphs, using GraphPad Prism^TM^ software (version 9) (San Diego, CA, USA). Taking into account the nonparametric distribution of the data, comparative analysis between groups were performed using the Mann–Whitney or Kruskal–Wallis tests, with significance of *p* < 0.05. The accuracy of the predictor was determined by the area under the curve (AUC) of the receiver operating characteristic (ROC). The AUCs, with 95% confidence intervals, were calculated to assess the diagnostic value of sTREM-1; AUC > 0.70 was considered clinically relevant. The dependence on multiple variables was calculated using Spearman’s correlation test, and the differences were considered statistically significant with *p* < 0.05. The calculated correlation matrix was presented graphically using the R package qgraph [24], and the principal component analysis (PCA) with PCATools [25]. Correlation coefficients (r) and *p*-values of the correlation matrix were formatted and tabulated, as seen in Appendix A. A ROC curve was applied to select the cutoff value for the sTREM-1 level to the best classification of death as an event. A multivariate Poisson regression model with robust variance was applied to adjust the incidence rate ratio (IRR) of deaths for the main risk variables such as sex, age, comorbidities, disease severity, days of disease, and laboratory markers, performed using STATA 15 (Stata Corp, College Station, TX, USA).

### 2.9. Ethical Approval

The procedures performed in this study were approved by the institutional ethics board of *Faculdade de Ciências Farmacêuticas de Ribeirão Preto–Universidade de São Paulo* and Brazil National Ethics Committee (CAAE: 30525920.7.0000.5403). Written informed consent was obtained from the participants.

## 3. Results

### 3.1. Demographic Data, Clinical and Laboratory Characteristics

In total, 237 patients with laboratory-confirmed COVID-19 were recruited for this study, categorized into residential care (60 subjects) and hospital care (177 subjects), in addition to 50 healthy volunteers (controls). The mean age was 35 years for healthy volunteers and 57 years for COVID-19 patients. The mean age of hospitalized patients was higher than that of domiciliary patients. The number of comorbidities was higher among patients with COVID-19 compared to controls, and among hospitalized patients compared to those with home care (Table 1). The most common initial symptoms in patients were cough, dyspnea, and dysgeusia, followed by diarrhea, fever, muscle soreness, and hyperactive delirium. Moreover, the absolute counts of erythrocytes, hemoglobin, neutrophils, lymphocytes, and monocytes in the COVID-19 patient group were different from those in the control group (Table 1).

The median time of hospitalization for COVID-19 patients was 9 days, and almost half of them required intensive care (46.3%). Additionally, these hospitalized patients received the following respiratory support: nasal-cannula oxygen (36.7%), oxygen mask ventilation (16.9%), and invasive mechanical ventilation (39.5%). Oxygen saturation was significantly lower among hospitalized patients compared to residential care patients upon evaluation of our investigations (Table 1). The most common pharmacological treatments for the patients with COVID-19 were glucocorticoid, azithromycin, ceftriaxone, oseltamivir, and colchicine, followed by chloroquine/hydroxychloroquine, anticoagulants, and ivermectin (Table 1).

### 3.2. Significant Association between sTREM-1 Release and Disease Severity

We compared plasma levels of sTREM-1 in control patients and COVID-19, further determining their correlation with the severity of the illness. As shown in Figure 1A, the sTREM-1 levels were higher in patients with COVID-19, especially in hospitalized subjects, with the lowest levels in healthy controls, indicating that this mediator could be a consequence of the infection or be involved in the activation of the inflammatory response against SARS-CoV-2.

The diagnostic value of sTREM-1 in patients with COVID-19 was evaluated by ROC curves. The AUC for sTREM-1 levels in patients under residential care, which represented mainly the mild form of the disease, was close to the value considered clinically relevant (AUC = 0.53). However, the AUC of sTREM-1 levels in patients under hospital care, who presented the severe form of COVID-19, was higher than the value considered to be clinically relevant (AUC = 0.96), thus distinguishing the disease severity among patients with COVID-19 (Figure 1B).

The plotted relative frequency of patients with sTREM-1 showed a high number of events for low levels of sTREM-1 among patients under residential care (mean = 72.32 pg/mL), and a greater number of events for high sTREM-1 levels among hospitalized patients (mean = 303.5 pg/mL), considering an intersection phase between residential and hospitalized care patients with COVID-19 (Figure 1C). These results demonstrated that sTREM-1 could be considered a potential predictive marker of the severity of COVID-19.

### 3.3. sTREM-1 Levels Increase with Clinical Disease Severity and Correlate with Comorbidities and the Production of Inflammatory Mediators in Patients with COVID-19

Next, we performed a subgroup analysis of sTREM-1, based on different severities of the disease in admission care. In this case, Sat O2, lymphocyte and neutrophil counts, BMI, hypertension, and IL-1β, IL-6, IL-8, IL-10, IL-12, and TNF production, were identified as independent risk factors or markers of adverse outcomes in patients with COVID-19 (Figure 2A—Heat map). However, when these parameters of inflammation or comorbidities were evaluated in the absence of sTREM-1 values, we were unable to specifically distinguish the subgroups of patients with COVID-19, but we observed a tendency to cluster between patients with critical and severe versus patients with mild and moderate COVID-19, using PCA (Figure 2B—PCA graphic). 

As shown in Figure 2C, sTREM-1 levels in COVID-19 patients increased with higher-severity forms of COVID-19 and significantly segregated subgroups of patients, as well as compared to control values.

In Figure 2D, the correlation matrix for Spearman’s test showed the significant positive association among sTREM-1 levels with severity and critical form of the disease, neutrophil count, BMI, age, sex male, and levels of IL-1β, IL-6, IL-8, and TNF. However, we also observed a significant negative association between sTREM-1 levels with the mild and moderate form of the disease, Sat O_2_, and lymphocyte counts. A network based on these data was constructed, analyzed, and graphically represented (Figure 2D). The complete data on *r* and *p*-values of this correlation matrix are available in Appendix A.

### 3.4. Multivariate Regression Analysis for the Prediction of Death and the Incidence Rate Ratio Value of sTREM-1 for the Expectation of Mortality Risk in COVID-19 Patients

To evaluate longitudinal changes, sTREM-1 levels in COVID-19 patients were monitored regularly during hospitalization. A cut-off value for a TREM-1 level of 188.93 pg/mL with the best-case classification rate of death (66.44%), with 0.90 (90%) of sensitivity (95% CI 0.79–0.95) and 0.60 (60%) of specificity (95% CI 0.41–0.63) was calculated (Figure 3A). To better see the overall changes in sTREM-1, plasma concentrations at admission (baseline level), and outcome (death or alive) were compared (Figure 3B).

The calculated area under the curve (AUC = 0.75, 95% CI 0.65–0.82, *p* < 0.0001) from ROC showed that plasma levels of sTREM-1 achieved good discrimination ability for death (Figure 3C). Additionally, the calculated positive likelihood ratio (LR+) was 1.81, and the negative likelihood ratio (LR–) was 0.21. The Poisson multivariate regression model was calculated by classifying patients with high or low serum TREM-1 levels according to the cut-off value (Appendix A). High levels of sTREM-1 were the preeminent predictive variable of deaths (IRR 2.94, *p* = 0.003), even after adjusting the model for sex (IRR 1.05, *p* = 0.290), age over 60 years (IRR 2.05, *p* = 0.002), number of comorbidities (IRR 1.11, *p* = 0.069), severity score (IRR 1.65, *p* = 0.001), days of disease (IRR 1.01, *p* = 0.174), and the neutrophil/lymphocyte ratio (NLR) (IRR 0.99, *p* = 0.612) (Figure 3D).

In Appendix A, we demonstrated the various pharmacological therapies administered to the cohort of hospitalized COVID-19 patients. Although there are no approved treatments for COVID-19, antibiotics, antivirals, and anti-inflammatory drugs were generally administered to patients under hospital care in Brazil. A Venn diagram represented the combinatorial pharmacological approach for these patients. The changes in oxygen support levels from hospital admission to recovery or death are shown in Appendix A, and the details of hospital support or supportive therapies for patients with COVID-19 are outlined in Appendix A.

### 3.5. sTREM-1 Was Released from the Surface of Peripheral Blood Leukocytes and Was Correlated with MMP-8 Expression in the COVID-19 Patient

Next, we determined the expression of TREM-1 on the peripheral blood cell surface of patients with COVID-19 [mild (*n* = 15), moderate (*n* = 15), severe (*n* = 15), and critical (*n* = 11)] and healthy controls (*n* = 10). In Figure 4A, we observe the sequential gating of SSC-A versus FSC-A; SSC-A versus CD14 antibody staining patterns; leukocytes gated according to their side scatter; and SSC-A versus CD16 antibody staining patterns. The percentage of CD14^+^ cells decreased in the blood of COVID-19 patients compared to the control. Furthermore, the percentages of CD14^+^TREM-1^+^ were lower in those COVID-19 patients. Conversely, the percentage of CD14^−^CD16^+^ cells was significantly increased by the COVID-19 severity. The frequency of CD14^−^CD16^+^TREM-1^+^ also increased on blood cell surfaces from moderate to critical COVID-19 groups compared to control and mild disease. The amount of TREM-1 expression was obtained by quantification of MFI in both CD14^+^ and CD14^−^CD16^+^ leukocytes. Interestingly, the total expression of TREM-1 decreased significantly according to the severity of the disease in the population of these cells. For example, these data indicated that the improved release of sTREM-1 matches the reduction in TREM-1 expression on the surface of the membrane of monocytes and polymorphonuclear leukocytes.

The proteolytic cleavage of TREM-1 anchored to mature membranes could be influenced by metalloproteinase. The expression of MMP-8 in plasma from COVID-19 patients followed the same pattern of sTREM-1 levels with a significant increase in the moderate (*n* = 23), severe (*n* = 20), and critical (*n* = 28) COVID-19 groups, compared to controls (*n* = 11) and mild (*n* = 19) (Figure 4B). The correlation between MMP-8 and sTREM-1 by the Spearman rank test was highly significant (*r* = 0.8640, *p* < 0.0001; Figure 4C). However, the direct effect of MMP-8 on TREM-1 cleavage is difficult to confirm in our human model. A detailed scheme of a proposed TREM-1 mechanism during COVID-19 is shown in Figure 5.

## 4. Discussion

To date, the most significant predictors of COVID-19 disease severity are related to the activation or suppression of host immune response [26,27]. This study reported that plasma sTREM-1 concentration might serve as a predictive factor or biomarker for disease severity and mortality in a population of patients with COVID-19 in Ribeirão Preto, São Paulo State, Brazil. The plasma levels of sTREM-1 gradually increased in patients with mild to moderate, severe, and critical forms of COVID-19. In fact, ROC analysis confirmed that sTREM-1 served as an important predictor of poor disease progression in patients with COVID-19. 

TREM-1 amplifies the pro-inflammatory innate immune response, in synergy with Toll-like receptors, which recognize a wide range of bacterial, fungal, and viral components [28]. In addition to expression by neutrophils, monocytes, and macrophages, TREM-1 is also expressed by epithelial and endothelial cells [29]. Elevated sTREM-1 levels are indicative of acute and chronic conditions, including sepsis and pneumonia [30,31]. The enhanced inflammatory response in macrophages has been indicated as a mechanism by which TREM-1 signaling contributes to lung injury; therefore, inhibition of TREM-1 is a potential therapeutic strategy for neutrophil lung inflammation and ARDS [32]. Moreover, the levels in sepsis could reflect an essential immune dysfunction, where excessive cleavage of TREM-1 could contribute to immunosuppression and death during severe infection [33]. 

Since multiple microorganisms can cause severe infections, the broad prognostic value of sTREM-1 indicates its potential as a severity marker for all-cause febrile disease [34]. However, fever was not the most common sign among patients with COVID-19 in our cohort, especially in cases of mild and moderate disease. While the role of sTREM-1 in patients presenting severe COVID-19 remains unclear, a functional genomic analysis of PBMCs from individuals undergoing infection with enterovirus A71 (EV-A71) determined that activation of TREM-1 was correlated with clinical severity [35]. Interestingly, genetic variants within the gene encoding TREM-1 are associated with different levels of inflammation and with the development of sepsis [36] and severe malaria in which sTREM-1 levels were high [8].

The number of patients with COVID-19 has increased drastically worldwide, mainly due to the prevalence of more infective variants. In this context, the early identification of patients prone to developing severe COVID-19 is essential for the triage and subsequent interventions. Some modeling studies have shown that levels of IL-6 and CRP could be used as independent factors to predict the COVID-19 severity [37]. IL-6 is a multifunctional cytokine that exhibits a strong pro-inflammatory effect [38]. Other studies have suggested that predictive values of IL-10 and IL-6 should be preferentially evaluated for the early diagnosis of patients with severe forms of this disease [39]. IL-10 is highly abundant, especially during the adaptive immune response [40]. However, although certain studies have shown that plasma IL-6 and IL-10 could be used as factors to predict the progression of COVID-19, in our cohort, their values could not be used in isolation to distinguish between patient groups. IL-6 is also elevated in obesity [41] and cannot be specific for COVID-19 patient conditions. Patients with the severe form of COVID-19 also had increased leukocytosis, neutrophilia, lymphopenia, and thrombocytopenia compared to those with non-severe forms of this disease [42]. In this way, we observed a significant correlation between plasma sTREM-1 and immunological parameters in patients with severe COVID-19.

We also inferred that sTREM-1 could be an independent discriminator of disease severity, in addition to predicting the poor outcome in patients with COVID-19. Therefore, significantly elevated plasma sTREM-1 levels in patients with COVID-19 might be indicative of an excessive inflammatory response and may contribute to severe illness or even death. Furthermore, other authors suggested that plasma sTREM-1 concentrations are related to COVID-19 severity and can discriminate between survivors and non-survivors, and sTREM-1 and IL-6 concentrations can be used as screening tools to decide treatment in patients with COVID-19 [20,21]. Generally, the most common therapeutic options for viral infections are directed at blocking viral replication or modulating immune responses. Dexamethasone exerts broad-spectrum anti-inflammatory effects. Among patients hospitalized with COVID-19, dexamethasone treatment resulted in lower mortality [43]. We evaluated the longitudinal sTREM-1 detection in patient admission and outcome under hospital care. Independent of dexamethasone treatment, sTREM-1 levels increased in most cases after admission and were correlated with the death outcome, but patients treated with dexamethasone, which was more common, tended to stabilize sTREM-1 production. However, this phenomenon did not correlate with better recovery. This represents a point of no return with anti-inflammatory treatment in COVID-19, and sTREM-1 levels could indicate the inflammatory state. Likely, the beneficial effect of glucocorticoids in patients with severe forms of COVID-19 can be dependent on the dosage and time.

In order for sTREM-1 to be a reliable marker of disease, a better understanding of the cells relevant to its release and mechanisms of shedding in COVID-19 are required. Another study reports a matrix metalloproteinase cleavage site within the TREM-1 sequence and demonstrates an in vivo correlation between MMP-9 expression, the appearance of sTREM-1 in airway lavage, and neutrophil recruitment during pulmonary influenza infection [44]. Additionally, it has been suggested that sTREM-1 is generated by cleavage of membrane-bound TREM-1 from the cell surface, and the infection model recruits many neutrophils that express this receptor.

In this case, a decrease in sTREM-1 could reflect a reduction in recruited neutrophils rather than a decrease in sTREM-1 release [16]. We observed that the reduction in the expression of TREM-1 on the cell surface in CD14^+^ and CD16^+^ leukocytes from the blood of COVID-19 patients was accompanied by a concomitant increase in plasma levels of sTREM-1. Indeed, the plasma expression of MMP-8 correlated positively with sTREM-1 levels, specifically in the severe COVID-19 patient group. These findings strongly support the hypothesis that proteolytic cleavage of membrane-anchored TREM-1 by one or several MMP could be responsible for a sTREM-1 generation. CD14 is a lipopolysaccharide (LPS)-binding protein, which functions as an endotoxin receptor [45]. CD14 is strongly positive in monocytes and most tissue macrophages but is weakly expressed or negative in monoblasts, promonocytes, and other granulocytic precursors, but neutrophils and a small proportion of B-lymphocytes may weakly express CD14 [45]. In the blood of COVID-19 patients, we observed a greater association of surface expression of TREM-1 with CD14^−^CD16^+^ than CD14^+^ cells. In addition, neutrophils have been reported as an important source of sTREM-1 in infectious processes [46], and it has been reported that neutrophils produce several MMP after LPS challenge [47]. However, the role of neutrophils in the TREM-1 pathway during COVID-19 needs further study.

There are certain limitations to our work. First, in some cases, in addition to COVID-19, chronic diseases and secondary infection can contribute to increased plasma levels of sTREM-1; however, to our knowledge, there are no reliable studies relating sTREM-1 with obesity, hypertension, and diabetes, common comorbidities in participants in our study. Second, this study was limited by the sample size. Larger cohort studies are necessary to further confirm the prognostic effect in critically ill patients with COVID-19. Third, some patients with critical illnesses were not admitted to intensive care units, which had a negative impact on the outcomes of these patients.

## 5. Conclusions

Our results suggest that plasma sTREM-1 levels at hospital admission can be used satisfactorily for evaluating disease severity and predicting adverse outcomes in patients with COVID-19. We suggest the potential role of sTREM-1 in determining the clinical course of COVID-19 and their correlation with other pro-inflammatory parameters. Furthermore, we propose a mechanism for the release of sTREM-1 established by MMP-8 activity on the surface of blood leukocytes. In this way, sTREM-1 has emerged as a potential prognostic biomarker that can be easily detected in plasma samples of COVID-19 patients.

## Figures and Tables

**Figure 1 viruses-13-02521-f001:**
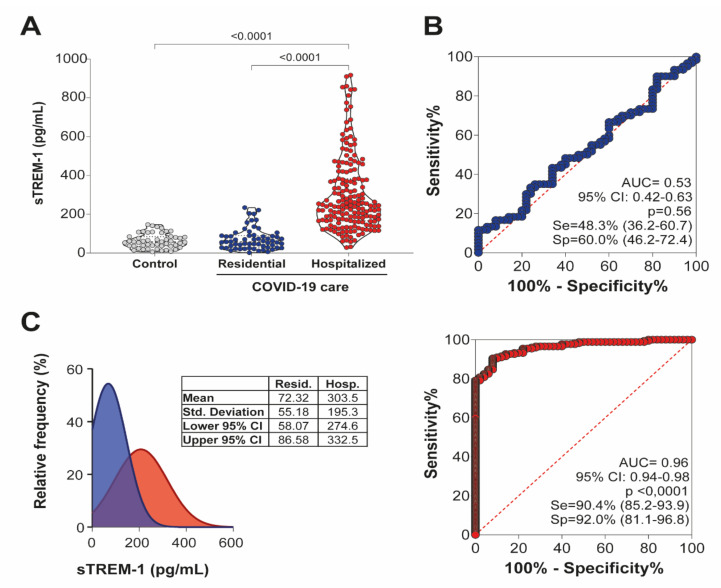
Elevated levels of sTREM-1 in patients with COVID-19. (**A**) The plasma concentration of sTREM-1 in COVID-19 patients under residential care (*n* = 60), hospital care (*n* = 177), or healthy controls (*n* = 50) were analyzed and compared. Data are presented as mean values plus ranges. The Kruskal–Wallis test was used to perform multiple comparisons when the data followed a non-normal distribution. The differences between the groups are indicated by the *p*-value in the graph above the diagram. (**B**) Receiver operating characteristic (ROC) curves of sTREM-1 concentrations to predict disease among patients with COVID-19 in residential care (blue) and hospital care (red). The area under the curve (AUC) and the *p*-values for significant differences between patients with COVID-19 and controls are depicted in the graphic. (C) Relative frequency of sTREM-1 between patients under residential care (blue) and hospital care (red) with COVID-19. Mean, standard deviation, lower 95% CI, and upper 95% CI data for each group are presented in the table between graphics.

**Figure 2 viruses-13-02521-f002:**
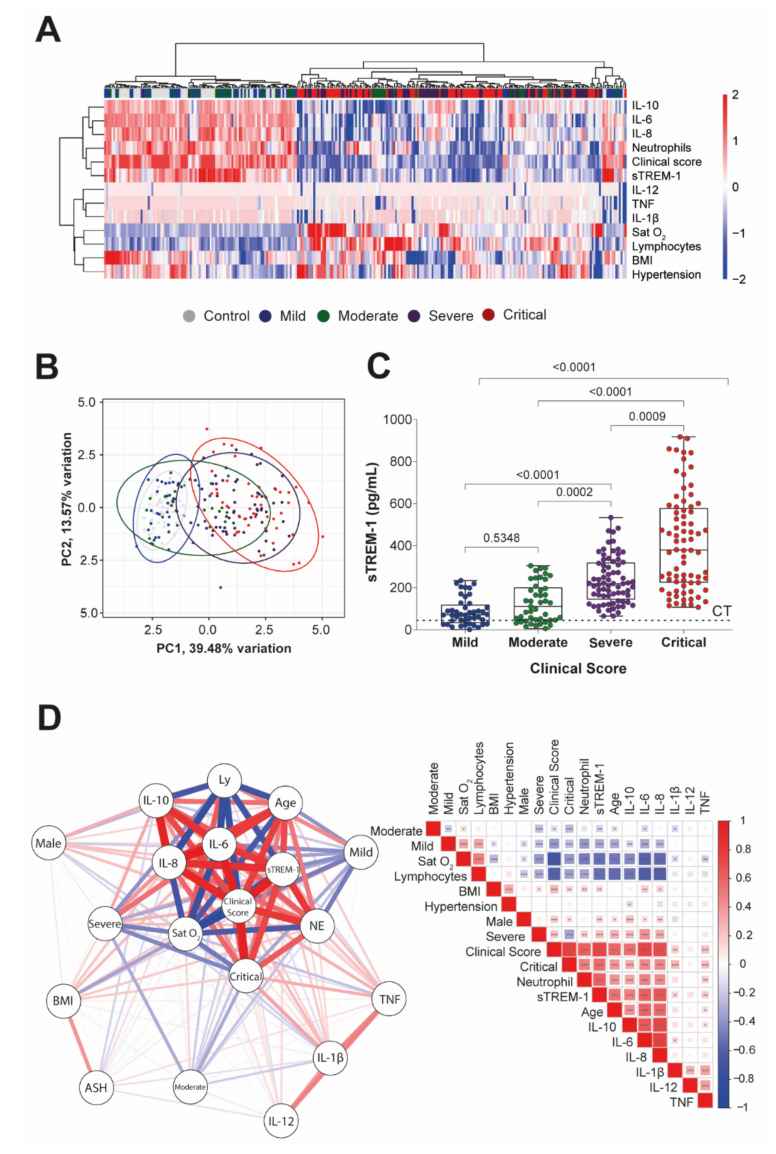
sTREM-1 levels were correlated with inflammatory cytokine production, comorbidities, and clinical severity of COVID-19 patients. (**A**) Unsupervised hierarchical cluster heat map showing the expression of markers and clinical parameters (Sat O_2_, lymphocytes (Ly), neutrophil (NE) counts, BMI, hypertension (ASH), sTREM-1, IL-1β, IL-6, IL-8, IL-10, IL-12, and TNF levels) for different groups of patients, according to the disease score (control, mild, moderate, severe, and critical) data are colored by row normalized value for each sample. (**B**) The PCA graphic shows the clusterization of those subgroups of patients (95% confidence interval), including all markers exhibited in the heat map, except for the levels of sTREM-1, performed with the base R functions. The continuous variables were transformed to log2 scale, and in the case of the heat map and PCA, the data used were transformed to z-scores (centered and scaled). (**C**) Levels of sTREM-1 in patients with different severity forms of COVID-19: mild (*n* = 44), moderate (*n* = 45), severe (*n* = 72), and critical (*n* = 76), as well as control (CT line, *n* = 50). Median values are presented with ranges. The Kruskal–Wallis test was used for multiple comparisons in data with non-normal distribution. The differences between each group are indicated by the *p*-value in the graphic above the diagram. (**D**) The color scale sidebar indicates the correlation coefficients (*r*), where red represents positive correlation, and blue represents negative correlation. The square size and color intensity are proportional to the correlation coefficients, *p* values were represented by * <0.05, ** <0.01, and *** <0.001. A network-based on Spearman’s correlation (*p* < 0.05) was constructed, analyzed, and graphically represented using the R packages.

**Figure 3 viruses-13-02521-f003:**
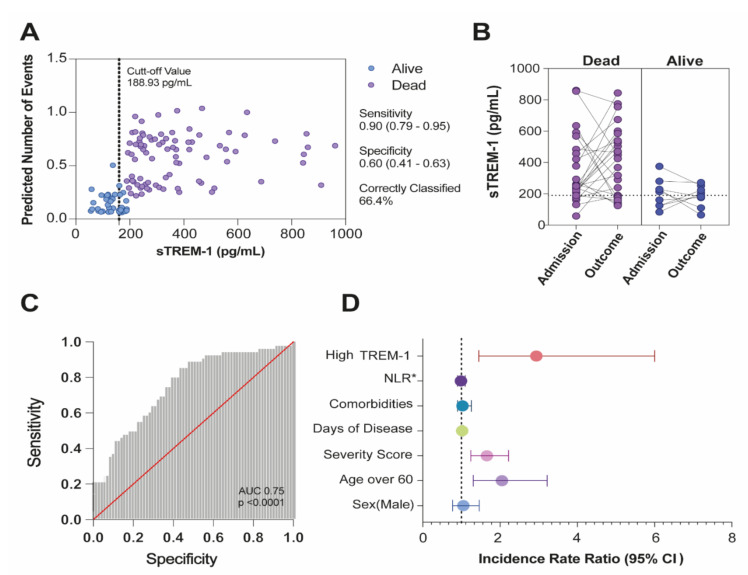
sTREM-1 levels in hospitalized patients with COVID-19 predicted mortality. (**A**) Adjusted predicted number of events for sTREM-1 levels. (**B**) Plasmatic levels of sTREM-1 of severe/critical patients were indicated for each individual on the day of hospital admission and after the outcome (dead or alive). The Mann–Whitney test was used for analyzing data with non-normal distribution, and differences between groups were established. (**C**) ROC and AUC assessing the discrimination capacity of sTREM-1 levels for mortality in hospitalized patients with COVID-19. (**D**) High levels of sTREM-1 and adjusted incidence rate ratio for the predictive variable of deaths. * Neutrophil/lymphocyte ratio (NLR).

**Figure 4 viruses-13-02521-f004:**
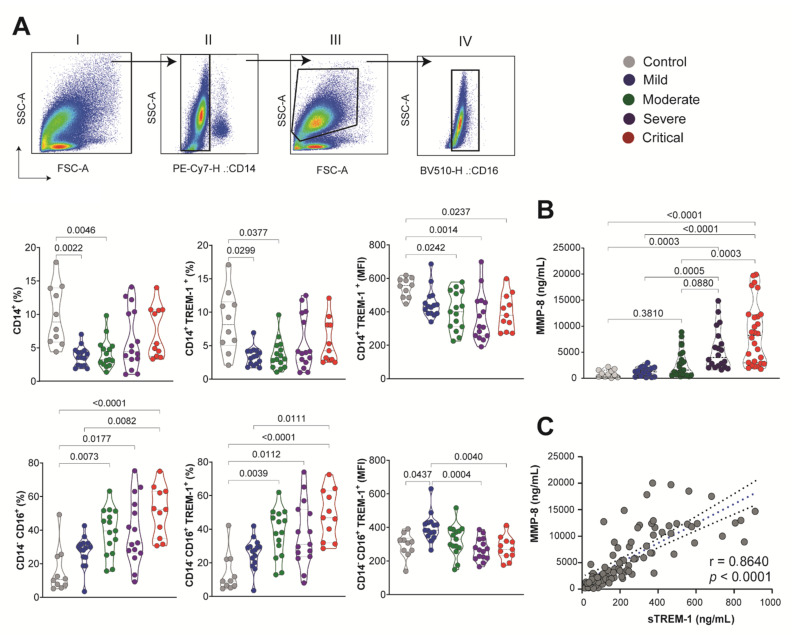
sTREM-1 was released from the surface of the peripheral blood leukocyte membrane, correlated with the expressionof MMP-8. (**A**) Sequential gating is shown: (I) SSC-A versus FSC-A; (II) leukocytes gated according to their side scatter and CD14 antibody staining patterns; (III) light scatter flow cytometry profile for cells based on forward scatter (FSC-A) related to size, and side scatters (SSC-A) related to granularity; (IV) gated according to their side scatter and CD16 antibody staining patterns. The percentage of CD14^+^ and CD14^−^ CD16^+^ cells was evaluated for groups of COVID-19 patients, as well as the percentage of CD14^+^TREM-1^+^ and CD14^−^CD16^+^TREM-1^+^ in the cell surface. The amount of TREM-1 expression was obtained by quantification of MFI in CD14^+^ and CD14^−^CD16^+^ leukocytes. (**B**) MMP-8 quantification in subgroups of COVID-19 patients. Median values are presented with ranges. The Kruskal–Wallis test was used for multiple comparisons in data with non-normal distribution. Differences between groups are indicated by the *p*-values in the graphics. (**C**) Spearman test correlation between MMP-8 and sTREM-1 levels. The correlation coefficients (r) and the *p*-value are indicated in the graphic.

**Figure 5 viruses-13-02521-f005:**
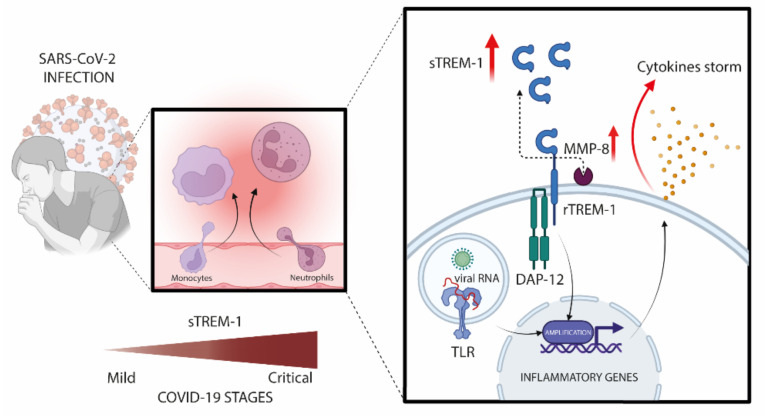
Schematic representation of the TREM-1/sTREM-1 pathway during the severity of COVID-19. SARS-CoV-2 activates innate immune receptors in infected cells and triggers the inflammatory transcription factor into the nucleus, where it induces several pro-inflammatory genes, including up-regulation of TREM-1. After binding to its ligand, TREM-1 associates with the adapter molecule DAP12, leading to a cascade of phosphorylation in a downstream kinase panel that, in turn, activates other transcription factors and amplifies the inflammatory response, as well as production of cytokines. It is speculated that high levels of circulating sTREM-1 were released from the surface of the peripheral blood leukocyte membrane related to MMP-8 activity. Therefore, individuals with high levels of sTREM-1 can indicate a dysregulated immune response (Created with BioRender.com, Agreement number: AZ238AM20F).

**Table 1 viruses-13-02521-t001:** Clinical and demographic data of COVID-19 patients enrolled in this study.

Baseline Variable	Healthy Controls N = 50	All Patients N = 237	COVID-19 Care		*p*-Value
			Residential N = 60	Hospitalized N = 177	
**Demographic characteristics**					
Age mean ± SD, (IQR)	35 ± 14.7 (19–80)	57 ± 19 (16–96)	37 ± 12 (16–71)	63 ± 16.4 (20–96)	**^a^ <0.0001** **^c,d^ <0.0001**
**Sex, No. (%)**					
Male	22 (44)	84 (35.4)	21 (35)	63 (36)	**^d^ 0.006**
Female	28 (56)	153 (64.6)	39 (65)	114 (64)	
**Comorbidities or coexisting disorders, No. (%)**					
Hypertension	7 (14)	116 (48.9)	4 (3.4)	112 (96.6)	**^a,c,d^ <0.0001**
Cardiovascular diseases	5 (18.5)	22 (81.5)	8 (36.4)	14 (63.6)	^a^ 0.7947 ^b^ 0.7685 ^c^ 0.5760 ^d^ 0.2084
Diabetes mellitus	4 (5.8)	65 (94.2)	5 (7.7)	60 (92.3)	**^a^ 0.0031** **^c^ 0.0002** **^d^ <0.0001**
History of smoking	3 (10)	27 (90)	6 (22.2)	21 (77.8)	^a^ 0.2189 ^b^ >0.1 ^c^ 0.1227 ^d^ 0.0692
History of stroke	-	10 (4.2)	-	10 (5.6)	-
Neurological diseases	-	12 (5.0)	2 (16.7)	10 (83.3)	^d^ 0.7353
Cancer	-	7 (2.9)	-	7 (3.9)	-
BMI (kg/m^2^)	26.5 ± 5.2 (15.4–43.2)	28.4 ± 7.0 (15.7–65.7)	27.2 ± 5.6 (17.7–43.8)	29.4 ± 7.3 (18.4–65.7)	**^a^ 0.0053** **^c^ 0.0008** **^d^ 0.0480**
**Presenting symptoms, No. (%)**					
Dyspnea	-	137 (57.8)	19 (31.6)	118 (66.6)	**^a,b,c,d^ <0.0001**
Fever	-	78 (32.9)	1 (1.7)	77 (43.5)	**^a,c,d^ <0.0001**
Myalgia	-	52 (21.9)	-	52 (29.4)	-
Diarrhea	-	56 (23.6)	22 (36.7)	34 (19.21)	**^d^ 0.0082**
Cough	-	161 (67.9)	43 (71.7)	118 (66.7)	**^d^ <0.0001**
Hyperactive delirium	-	15 (6.3)	-	15 (8.5)	-
Dysgeusia	-	60 (25.3)	40 (66.7)	20 (11.3)	**^d^ <0.0001**
Anosmia	-	67 (28.3)	41 (68.3)	26 (14.7)	**^d^ <0.0001**
**Laboratory findings, mean ± SD, (IQR)**					
Erythrocytes × 10^9^/L	4.6 ± 0.6 (3.6–5.8)	4.4 ± 0.8 (2.0–5.9)	4.8 ± 0.4 (3.7–5.8)	4.2 ± 0.8 (2.0–5.9)	**^a^ 0.0078** **^c,d^ <0.0001**
Hemoglobin (g/dL)	14.5 ± 1.6 (10.5–17.5)	13.1 ± 2.6 (6.6–18.2)	14.5 ± 1.3 (12.0–17.7)	12.4 ± 2.6 (6.6–18.2)	**^a,c,d^ <0.0001**
Leukocytes × 10^9^/L	7.5 ± 1.8 (4.1–13.2)	9.0 ± 5.6 (1.6–33)	7.3 ± 2.1 (3.2–13.5)	10.2 ± 6.0 (1.6–33)	**^c,d^ <0.0001** **^a^ 0.0088**
Neutrophils × 10^9^/L	4.2 ± 1.4 (2.4–9.8)	7.0 ± 5.0 (1.4–26.1)	4.1 ± 1.9 (1.6–11.0)	8.3 ± 5.0 (1.4–26.1)	**^a,c,d^ <0.0001**
Lymphocytes × 10^9^/L	1.2 ± 0.7 (1.2–5.0)	1.3 ± 0.9 (0.1–4.2)	2.3 ± 0.6 (1.1–4.3)	1.0 ± 0.7 (0.1–4.1)	**^a,c,d^ <0.0001**
RNL	1.8 ± 1.0 (0.9–6.3)	5.6 ± 6.4 (0.1–30.7)	1.7 ± 1.0 (0.5–7.9)	7.3 ± 6.4 (0.2–30.7)	**^a,c,d^ <0.0001**
Monocytes × 10^9^/L	0.5 ± 0.2 (0.1–1.4)	0.5 ± 0.3 (0.1–1.9)	0.5 ± 0.1 (0.2–0.9)	0.5 ± 0.4 (0.1–1.9)	^a,b,c,d^ >0.1
Platelets × 10^9^/L	214 ± 51.8 (129–370)	244 ± 98.1 (50–635)	227 ±66.3 (119–474)	245 ± 105.8 (50–635)	^a,b,d^ >0.1 ^c^ 0.0775
**Hospital support, No. (%)**					
Infirmary	-	95 (40)	-	95 (53.7)	-
Intensive care unit (ICU)	-	82 (34.6)	-	82 (46.3)	-
**Hospitalization data, No.**					
Hospitalization days, mean (IQR)	-	9 (1–30)	-	9 (1–30)	-
**Respiratory support upon assessment (%)**					
Nasal-cannula oxygen	-	65 (27.4)	-	65 (36.7)	-
Oxygen masks/noninvasive	-	30 (12.6)	-	30 (16.9)	-
Invasive mechanical ventilation	-	70 (29.5)	-	70 (39.5)	-
Oxygen saturation mean ± SD (IQR)	99 ± 2.4 (89–99)	93 ± 8.7 (54–99)	98 ± 1.8 (92–99)	91 ± 8.9 (54–99)	**^a,c,d^ <0.0001**
**Medications No. (%)**					
Glucocorticoid	-	156 (61.6)	10 (16.7)	146 (82.5)	**^d^ <0.0001**
Azithromycin	-	149 (68.6)	18 (30.0)	127 (71.7)	**^d^ <0.0001**
Ceftriaxone	-	84 (33.7)	0.4 (6.7)	80 (45.2)	**^d^ <0.0001**
Oseltamivir	-	80 (30.4)	0.8 (13.3)	72 (40.7)	**^d^ <0.0001**
Colchicine	-	0.5 (2.1)	-	0.5 (2.8)	-
Chloroquine/hydroxychloroquine	-	18 (7.6)	-	26 (14.7)	-
Anticoagulant	-	34 (14.3)	-	34 (19.2)	-
Ivermectin	-	11 (4.6)	11 (18.3)	-	-

^a^ Comparisons between healthy controls and COVID-19 patients; ^b^ healthy controls versus non-hospitalized COVID-19 patients; ^c^ healthy controls versus hospitalized COVID-19 patients; ^d^ non-hospitalized versus hospitalized COVID-19 patients. Patient data were compared using the chi-square test, or Fisher’s exact test for categorical variables and one-way analysis of variance (ANOVA). Mann–Whitney, nonparametric t-test was used for continuous variables. *p* < 0.05 was considered statistically significant. Abbreviations: standard deviation (SD); data are median (IQR), *n* (%), or n/N.

## Data Availability

All data generated or analyzed during this study are included in this published article and its preprint manuscript form: https://doi.org/10.1101/2020.09.22.20199703.

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
