# Peer review of "sTREM-1 Predicts Disease Severity and Mortality in COVID-19 Patients: Involvement of Peripheral Blood Leukocytes and MMP-8 Activity"

_viruses, 2021, doi:10.3390/v13122521_

Round 1

Reviewer 1 Report

This study tries to use sTREM-1 as a predictive tool of disease outcomes (severity or death) of COVID-19 for improved management.

In the introduction 2 important paper investigating the same role of sTREM-1 in COVID-19 patients could not be found: Van Singer M, Brahier T, Ngai M, Wright J, Weckman AM, Erice C, Meuwly JY, Hugli O, Kain KC, Boillat-Blanco N. COVID-19 risk stratification algorithms based on sTREM-1 and IL-6 in emergency department. J Allergy Clin Immunol. 2021 Jan;147(1):99-106.e4. doi: 10.1016/j.jaci.2020.10.001 and de Nooijer AH, Grondman I, Lambden S, Kooistra EJ, Janssen NAF, Kox M, Pickkers P, Joosten LAB, van de Veerdonk FL, Derive M, Gibot S, Netea MG; RCI-COVID-19 study group. Increased sTREM-1 plasma concentrations are associated with poor clinical outcomes in patients with COVID-19. Biosci Rep. 2021 Jul 30;41(7):BSR20210940.

The description of the results of these studies may largely tone down the conclusions of this study and give the authors the opportunity perhaps to highlight the differences.

In the patient inclusion it is difficult to understand why there is a significant difference in the age of the controls and the diseased subjects. Why the authors did not choose an age-matched control group.

It is difficult also to understand why the authors did not include other inflammatory markers such as the TNFalpha.

In the flow cytometry section why the authors did not use direct neutrophils markers such as the CD66 or CD16 or CD11b. The use of CD14- state decrease highly the validity of the source of sTREM-1 shedding.

The results section is adequate.

In the discussion one would expect that the authors make a mechanistic description of the eventual role of sTREM-1 for determining the severity trajectory of the COVID-19 patients. The conclusion is highly descriptive and generalized.

It is also interesting that in Figure2B there is a great overlap between the values especially between the severe and critical patients. What does it mean? What differentiates a Severe or Critical patient with the same sTREM-1 value. The same could be said between the moderate and severe cases.

Author Response

Reviewer #1

This study tries to use sTREM-1 as a predictive tool of disease outcomes (severity or death) of COVID-19 for improved management. In the introduction 2 important paper investigating the same role of sTREM-1 in COVID-19 patients could not be found: Van Singer M, Brahier T, Ngai M, Wright J, Weckman AM, Erice C, Meuwly JY, Hugli O, Kain KC, Boillat-Blanco N. COVID-19 risk stratification algorithms based on sTREM-1 and IL-6 in emergency department. J Allergy Clin Immunol. 2021 Jan;147(1):99-106.e4. doi: 10.1016/j.jaci.2020.10.001 and de Nooijer AH, Grondman I, Lambden S, Kooistra EJ, Janssen NAF, Kox M, Pickkers P, Joosten LAB, van de Veerdonk FL, Derive M, Gibot S, Netea MG; RCI-COVID-19 study group. Increased sTREM-1 plasma concentrations are associated with poor clinical outcomes in patients with COVID-19. Biosci Rep. 2021 Jul 30;41(7):BSR20210940.

Answer: In fact, we emphasize that our data on the non-profit preprint server MedRxiv.org (doi.org/10.1101/2020.09.22.20199703) mentioned for the first time the correlation of sTREM-1 levels with COVID-19 severity and its potential use as a biomarker. However, we agree with the reviewer to reference the other works, and we have revised this point accordingly in the introduction and discussion section.

The description of the results of these studies may largely tone down the conclusions of this study and give the authors the opportunity perhaps to highlight the differences.

Answer: We have revised this point accordingly.

In the patient inclusion it is difficult to understand why there is a significant difference in the age of the controls and the diseased subjects. Why the authors did not choose an age-matched control group.

Answer: This is a really interesting and important question. We had intended to have an age-matched control group with patients, but the context of the pandemic and the health recommendations of social isolation discouraged the participation and exposure of older persons as volunteers. Even though the mean age of our healthy controls was 35, we enrolled volunteers from 19 to 80 years in this group. This circumstance is likewise observed in other works with the COVID-19 subject. However, through Spearman's correlation test and Multivariate Poisson Regression model, we identified that the age factor is not a confounding factor for the sTREM-1 release and our data has been validated.

It is difficult also to understand why the authors did not include other inflammatory markers such as the TNF-alpha.

Answer: We thank the reviewer for this important suggestion, and we include new data from other inflammatory markers such as IL-8, IL-1beta, IL-12, and TNF in Figure 2. However, it was observed that the production of cytokines IL-1beta, IL-12, and TNF was not robust in the plasma of COVID-19 patients.

In the flow cytometry section why the authors did not use direct neutrophils markers such as the CD66 or CD16 or CD11b. The use of CD14- state decrease highly the validity of the source of sTREM-1 shedding

Answer: It is a very relevant comment. This approach, while often providing important information, does not provide a complete picture of immune cell repertoires and does not allow the examination of dynamic changes in all immune cell populations. At the beginning of the study, our main focus was not to evaluate the relationship between neutrophils and TREM-1, rather than mononuclear cells. Then we did not include specific antibodies to stain this population, except for anti-CD16, which is also used to discriminate monocyte populations. We agree that in light of the present results, it would be important to accomplish our conclusions. However, due to temporary pandemic control in our region and a high number of vaccinated people, it is not possible, at this moment, to enroll additional participants in our study, that should have the same clinical and immunological characteristics as those previously found. Nevertheless, in an attempt to partially answer this question, we re-analyzed the flow cytometry data, by selecting CD14 negative population according to size and granularity parameters, compatible with polymorphonuclear cells. Following a further gating in CD16 + cells, we estimated the frequency of sTREM-1 expressed by this population, which could typically comprise neutrophils. These results were added to the revised version of the manuscript in Figure 4.

The results section is adequate.

Answer: The authors thank Reviewer #1 for the positive evaluation of our manuscript.

In the discussion one would expect that the authors make a mechanistic description of the eventual role of sTREM-1 for determining the severity trajectory of the COVID-19 patients. The conclusion is highly descriptive and generalized.

Answer: We have revised this point accordingly in the Discussion and Conclusion sections.

It is also interesting that in Figure2B there is a great overlap between the values, especially between the severe and critical patients. What does it mean? What differentiates a Severe or Critical patient with the same sTREM-1 value. The same could be said between moderate and severe cases.

Answer: We respected the reviewer’s opinion, but we consider that those patients were stratified by clinical parameters, which diverge according to the medical team observation and management applied to patients. Furthermore, the evolution of patients regarding symptoms is very dynamic and depends on several factors, such as response to pharmacological treatment and the presence of comorbidities or other secondary infections. Thus, several inflammatory biomarkers overlap in this scenario, but the mean values of sTREM-1 followed the severity of COVID-19 regardless of whether some patients had a range that overlapped with other groups.

Reviewer 2 Report

The authors have presented a very interesting study that shows promising results about the potential of sTREM-1 as a predictor of COVID-19 severity and mortality. 

The study results are very well presented and described clear and adequately. 

i have only 2 minor questions: 

could the authors present Figure 2A in unsupervised clustering according to row and column? do the groups actually cluster together?

and Fig. 2 C: which of the correlations is significant? 

Author Response

Reviewer #2

The authors have presented a very interesting study that shows promising results about the potential of sTREM-1 as a predictor of COVID-19 severity and mortality.

The study results are very well presented and described clear and adequately.

Answer: The authors thank reviewer #2 for the positive evaluation of our manuscript.

I have only 2 minor questions:

could the authors present Figure 2A in unsupervised clustering according to row and column? do the groups actually cluster together?

Answer: We thank the reviewer for this important question, and we have revised this point accordingly. In figure 2A we replaced the supervised Heat map for an unsupervised analysis, thus demonstrating that most individuals were clustered into two main groups involving mild/moderate and severe/critical.

and Fig. 2 C: which of the correlations is significant?

Answer: We apologize for not being clear enough. However, Spearman's correlation values r and p were shown in a supplemental table (supplementary Table 2), but we now added characters (*) to Figure 2C to represent statistical significance.

Reviewer 3 Report

I read with interest the manuscript by da Silva-Neto and colleagues on the possibility to use sTREM-1 as a marker of severity of SARS-CoV-2 infection. I believe that regardless of some critical issues, the study is of interest to the scientific community. There are very little data on the topic and in my opinion it is worth further investigation.
These are my comments:
- Why did the authors not follow the classification of symptoms disclosed by the WHO?
- When was the blood draw performed with respect to the onset of symptoms?
- In the choice of the 50 control subjects, the authors have not taken into account the differences in age compared to the group of patients. It would be useful to expand the number of healthy subjects to better represent the age range of patients.
- Although there are very few studies on the association of TREM-1 and COVID-19, these should be cited.
- I do not think it is so much the blood glucose level as the diabetes that is significant of the disease course, also the authors should also take into account the dexamethasone treatment. 

Author Response

Reviewer #3

I read with interest the manuscript by da Silva-Neto and colleagues on the possibility to use sTREM-1 as a marker of severity of SARS-CoV-2 infection. I believe that regardless of some critical issues, the study is of interest to the scientific community. There are very little data on the topic and in my opinion it is worth further investigation.

Answer: The authors thank Reviewer #3 for the positive evaluation of our manuscript.

These are my comments:

- Why did the authors not follow the classification of symptoms disclosed by the WHO?

Answer: In fact, at the beginning of this work, little information was available on the classification of the COVID-19 patient. In this way, we follow the most mentioned works referenced by other studies and also the basis for the WHO protocol. However, we made some adaptations regarding the reality of the management and clinical information of patients treated in the Brazilian health system. Throughout the pandemic, some updates were proposed by WHO, but we chose to keep on our initial experimental design, and to clarify our parameters for patient classification we added supplementary information (Supplementary Table 1). References: Hadjadj, J.; Yatim, N.; Barnabei, L.; Corneau, A.; Boussier, J.; Smith, N.; Péré, H.; Charbit, B.; Bondet, V.; Chenevier-Gobeaux, C.; et al. Impaired Type I Interferon Activity and Inflammatory Responses in Severe COVID-19 Patients; Ye, G.; Pan, Z.; Pan, Y.; Deng, Q.; Chen, L.; Li, J.; Li, Y.; Wang, X. Clinical Characteristics of Severe Acute Respiratory Syndrome Coronavirus 2 Reactivation. Journal of Infection 2020, 80, doi:10.1016/j.jinf.2020.03.001. Xu, X.W.; Wu, X.X.; Jiang, X.G.; Xu, K.J.; Ying, L.J.; Ma, C.L.; Li, S.B.; Wang, H.Y.; Zhang, S.; Gao, H.N.; et al. Clinical Findings in a Group of Patients Infected with the 2019 Novel Coronavirus (SARS-Cov-2) Outside of Wuhan, China: Retrospective Case Series. The BMJ 2020, 368, doi:10.1136/bmj.m606. Grasselli, G.; Zangrillo, A.; Zanella, A.; Antonelli, M.; Cabrini, L.; Castelli, A.; Cereda, D.; Coluccello, A.; Foti, G.; Fumagalli, R.; et al. Baseline Characteristics and Outcomes of 1591 Patients Infected with SARS-CoV-2 Admitted to ICUs of the Lombardy Region, Italy. JAMA - Journal of the American Medical Association 2020, 323, 1574–1581, doi:10.1001/jama.2020.5394. Marshall, J.C.; Murthy, S.; Diaz, J.; Adhikari, N.; Angus, D.C.; Arabi, Y.M.; Baillie, K.; Bauer, M.; Berry, S.; Blackwood, B.; et al. A Minimal Common Outcome Measure Set for COVID-19 Clinical Research. The Lancet Infectious Diseases 2020, Office, W.H.O.E.M.R. Updated Clinical Management Guideline for COVID-19. Weekly Epidemiology Monitor 2020; Wei, P.-F. Diagnosis and Treatment Protocol for Novel Coronavirus Pneumonia (Trial Version 7). Chinese Medical Journal 2020, 133, 1087–1095, doi:10.1097/CM9.0000000000000819. Wan, S.; Xiang, Y.; Fang, W.; Zheng, Y.; Li, B.; Hu, Y.; Lang, C.; Huang, D.; Sun, Q.; Xiong, Y.; et al. Clinical Features and Treatment of COVID-19 Patients in Northeast Chongqing. Journal of Medical Virology 2020, 92, 797–806, doi:10.1002/jmv.25783.

 - When was the blood draw performed with respect to the onset of symptoms?

Answer: We apologize for the omission of this information. We have included a new statement in the Materials and Methods section. By the way, the patients’ blood was collected approximately 6.1 ± 2.8 days after symptom onset. As this was an opportunity sample, we decided not to control this variable.

 - In the choice of the 50 control subjects, the authors have not taken into account the age differences compared to the group of patients. It would be useful to expand the number of healthy subjects to better represent the age range of patients

Answer: This is a really interesting point. We had intended to have an age-matched control group with patients, but due to the context of the pandemic and the health recommendations of social isolation discouraged the participation and exposure of older persons as volunteers. Even though the mean age of our healthy controls was 35, we enrolled volunteers from 19 to 80 years in this group. This circumstance is likewise observed in other works with the COVID-19 subject. However, through Spearman's correlation test and Multivariate Poisson Regression model, we identified that the age factor is not a confounding factor for the sTREM-1 release and our data has been validated.

- Although there are very few studies on the association of TREM-1 and COVID-19, these should be cited.

Answer: In fact, we emphasize that our data in the not-for-profit preprint server MedRxiv.org (doi.org/10.1101/2020.09.22.20199703) mentioned for the first time the correlation of sTREM-1 levels with COVID-19 severity and its potential use as a biomarker. However, we agree with the reviewer to reference the others works and we have revised this point accordingly in the introduction and discussion section.

- I do not think it is so much the blood glucose level as the diabetes that is significant of the disease course, also the authors should also take into account the dexamethasone treatment.

Answer: The reviewer is right, and we modified this statement in the text as suggested.

Round 2

Reviewer 1 Report

The authors adequately addressed all my concerns.